# OpenReview forum: "INDICT: Code Generation with Internal Dialogues of Critiques for Both Security and Helpfulness"
_NeurIPS.cc/2024/Conference — NeurIPS 2024 poster_

### Official Review · Reviewer_ZHvf · 2024-06-26

**Soundness:** 2
**Presentation:** 3
**Contribution:** 2
**Rating:** 5
**Confidence:** 5

**Summary:**

This paper introduces a dual-critic prompting framework, INDICT, to consider both helpfulness and security during code generation. Specifically, the author introduces two critics, one for helpfulness and the other for security, to provide suggestions on improving the initially generated code. Besides, the authors equipped two critics with external tools like search engines and code executors to address the hallucination problem and employ an iterative refining style to further improve the performance. The authors evaluate INDICT on 8 tasks across 5 benchmarks on several LLMs. The experimental results demonstrate the effectiveness of INDICT.

**Strengths:**

1. The paper is well-written and easy to follow.
2. The studied topic is valuable. According to authors' discussion of related works, it seems that there are many works focusing on pointing out the security problems of LLM-generated codes, yet the papers for solutions are few.
3. The ablation studies demonstrate the effectiveness of each module (helpfulness critic, security critic, external tools, iterative rounds).

**Weaknesses:**

1. Though no former works using a multi-agent collaborative system addressing the security of code generation according to related works part in the paper, it is generally believed and widely proven effective of adopting a multi-agent collaborative system during content generation [1,2,3], which weakens the novelty of the paper.
2. The experiments only compare INDICT with pure LLMs. According to the related work part, there are several related works. Although the authors mentioned that related works are either not adapted for benchmarks used in this paper or not accessible, a lack of comparison with related works weakens the validity of the effectiveness of INDICT.
3. The authors discussed an alternative by using only one LLM to act as a helpfulness critic and security critic at the same time in their method description section. However, I do not find any experiments comparing using two LLMs as critics and using only one. The lack of this comparison confuses me with the necessity of employing two LLMs separately.

[1] Dong, Yihong, et al. "Self-collaboration code generation via chatgpt." arXiv preprint arXiv:2304.07590 (2023).

[2] Huang, Dong, et al. "Agentcoder: Multi-agent-based code generation with iterative testing and optimisation." arXiv preprint arXiv:2312.13010 (2023).

[3] Ishibashi, Yoichi, and Yoshimasa Nishimura. "Self-organized agents: A llm multi-agent framework toward ultra large-scale code generation and optimization." arXiv preprint arXiv:2404.02183 (2024).

**Questions:**

1. What is the meaning of 'GT Safe' and 'GT Unsafe' in Fig.4 (c)?
2. Can you provide the time cost of INDICT? Since your system contains multiple LLMs, multiple rounds of interaction, and even the usage of search engines and code executors, it is necessary for readers to know the efficiency trade-off compared with pure LLMs. Note that I am not taking this point as a weakness of INDICT.

**Limitations:**

The authors discussed limitations in Appendix A. Among the discussed limitations, I think the last one needs further discussion. Though the authors discussed that INDICT is much more efficient compared with fine-tuning LLMs that require curated training examples, once the fine-tuned LLM is released, the inference time costs much less than INDICT because INDICT is quite heavy.

---

> ### Author Rebuttal · Authors · 2024-08-06
>
> Thank you for your reviews! Please refer to our responses below.
>
> ### Q1: ..It is generally believed and widely proven effective of adopting a multi-agent collaborative system during content generation
> Even though multi-agent collaborative systems have been proposed for content generation, it is not trivial to extend this line of research to address the security of code generation.
> * Firstly, it is not clear how the current methods could be enhanced with security awareness and subsequently improve the quality of output generations. Earlier work such as [1, 2] showed that simply asking agents to analyze and answer what is wrong with generated code is not always effective. With carefully designed prompts and agent interactions [3, 4], collaborative agents can now generate more functionally correct code.
> * Therefore, studying collaborative agents with orthogonal goals such as security awareness still requires further attention. As observed by our additional experimental results (see our global response#2 above), simply applying the current multi-agent methods [5, 6], even with extended additional instructions of security criteria, does not perform so well and is still far from optimal.
>
> Furthermore, we would like to emphasize that our method novelty is not only about the multi-agent collaborative system. We also proposed an innovative multi-critic framework as an internal reasoning module with access to external tools for knowledge grounding. Our strategy can integrate holistic and reliable critic feedback from code execution outputs as well as supporting information snippets through multiple rounds of agent interactions. As demonstrated by our results (see our global response#2 above), our method is more well-rounded with high performance by both safety and helpfulness. Refer to our global response#1 above for a more comprehensive and systematic comparison between INDICT and other related work.
>
> ### Q2: The experiments only compare INDICT with pure LLMs.…
> Following your recommendations, we selected 7 strong baselines from related lines of research, including self-refine, multi-agent, and finetuning methods (see our global response#2 above). For most baselines, we also include a version of the method where additional instructions are given to the model to follow and provide both safety and helpfulness feedback e.g. we instructed models to “focus on both the security and helpfulness of the solution.” For Self-Collab [5], we included these instructions in both analyst and tester agents. For CAMEL [6], we follow the Critic-in-Loop setup as recommended in the paper appendix. Note that for all baseline models, we followed the same generation budgets to fairly compare the results (up to 3 rounds of revision).
>
> From the experiment results (see our global response#2 above), we can observe the SoTA performance of INDICT. While we observed the improvement of baseline methods with additional instructions (marked with the suffix ‘+’), their results are still not as good as INDICT in terms of helpfulness and safety. For finetuning method CodeUltraFeedback, their best model (SFT+DPO fine-tuned) is still far from perfect and could be further optimized. We further integrated the fine-tuned models with INDICT and observed a significant performance boost. We will include these results and more detailed analysis in our revised paper.
>
> ### Q3: The authors discussed an alternative by using only one LLM to act as a helpfulness critic and security critic at the same time...
> We conducted additional ablation analysis to evaluate whether one LLM can act as a critic for both helpfulness and safety at the same time. We combined the previous criteria for security and helpfulness and integrated them into the prompt for this critic agent. From the results (see our global response #3), simply using a single critic agent with dual quality criteria will affect the performance, reducing the safety and helpfulness metrics to 87% and 76% respectively. One possible reason is due to the formulation of the training objectives of LMs, which are not always designed to optimize both security and helpfulness equally (also depending on the post-pretraining stages of LMs e.g. training with RLHF). Our approach enables a more flexible and probably more relaxed application of LLM as a critic agent by:
> * (1) decoupling the helpfulness and safety goals and delegating them to individual LM agents; and
> * (2) enabling multi-critic collaboration to autonomously develop more holistic and well-rounded critic feedback.
>
> We will include the results and a more detailed analysis in our revised paper.
>
> ### Q4: What is the meaning of 'GT Safe' and 'GT Unsafe' in Fig.4 (c)?
> “GT Safe” and “GT Unsafe” denote the ground-truth secure and insecure code outputs provided by the CVS benchmark. We will explain and make the definitions clearer in our revised paper.
>
> ### Q5: ...Can you provide the time cost of INDICT? ... once the fine-tuned LLM is released, the inference time costs much less than INDICT
>
> On average, INDICT incurs about 3 to 4x the time cost as compared to pure LLMs. We also want to note that even with fine-tuning, fine-tuned models are far from perfect and still subject to unseen security risks or novel red-teaming prompts during test time. For instance, from our results with the fine-tuning method CodeUltraFeedback (see our global response#3 above), the fine-tuned model is still sub-optimal and can be further improved e.g. using INDICT during inference time (improving the performance from 60% to 73%).
>
> *[1] Is self-repair a silver bullet for code generation?*
>
> *[2] Large language models cannot self-correct reasoning yet*
>
> *[3] Reflexion: Language Agents with Verbal Reinforcement Learning*
>
> *[4] AgentCoder: Multiagent-Code Generation with Iterative Testing and Optimisation*
>
> *[5] Self-collaboration Code Generation via ChatGPT*
>
> *[6] CAMEL: Communicative Agents for "Mind" Exploration of Large Language Model Society*

---

> > ### Comment · Reviewer_ZHvf · 2024-08-10
> > **Response to authors**
> >
> > Thank you for your clarification. I decided to raise my score.
> > Please add this rebuttal content to the next version of the paper.
> >
> > One more thing, I still strongly recommend a more detailed time cost comparison between INDICT and pure LLMs, other baselines. To give readers better insights about the efficiency trade-off.

---

> > > ### Author Response · Authors · 2024-08-11
> > >
> > > Thank you for your consideration and revising the score! We will incorporate the feedback and our discussion in detail into the revised paper.
> > >
> > > Regards,

---

### Official Review · Reviewer_rV62 · 2024-07-08

**Soundness:** 3
**Presentation:** 3
**Contribution:** 2
**Rating:** 6
**Confidence:** 3

**Summary:**

The paper describes a method to improve the helpfulness and safety of LLMs for code generation tasks. It uses two critics - one for safety and one for helpfulness that communicate with each other to iteratively provide feedback to the actor model or the actual agent that is tasked with the code completion task. The critics are further augmented with tools that allow using the web for searches as well as LLMs from OpenAI and a code interpreter to execute code. The implementation of the method has been done using CommandR+ though experiments have also been presented using CodeLlama and Llama for certain tasks. Prompts have been standardized across LLMs to the extent possible. Experiments on benchmarks for Code Security (insecure code generation, malicious code generation) as well as open-ended generation tasks from HarmBench indicate the method improves the performance of  the corresponding base model. Ablation experiments on the code security tasks (done using CodeLlama and CommandR+ indicate that the use of both critics as well as tools help improve performance.

**Strengths:**

- Well written paper with a detailed appendix
- Simple but interesting extension of the use of collaborative agents to improve safety and helpfulness for code tasks
- Improved performance on multiple tasks

**Weaknesses:**

- Despite some details in the appendix; the evaluation section is missing crucial details
- See Questions

**Questions:**

- For the HarmBench evaluation - how have the red-teaming methods been applied with the critics in place? I'm guessing it was applied on the actor/agent prompt? Some details would be helpful.  Was the evaluation also done using completions?  The main text suggests so, but there are no tables/results referenced. Table 1 isn't discussed (no mention or reference of the red-teaming methods in the main paper). This evaluation section is a bit unclear to me.
- Since there are no samples of the generated dialogs or the system prompt for the actor (agent) -- if any; I'd imagine a well rounded summary being made available to the agent (Equation 5). Could the authors share more details/prompt along with a sample?
     * I was curious if fine-tuning models (smaller models?) on such generated data would have been a good experiment? Doing so could help in reducing inference cost by employing cheaper/smaller models

**Limitations:**

Yes for the most part.

---

> ### Author Rebuttal · Authors · 2024-08-07
>
> Thank you for your comments. Please refer to our responses below.
>
> ### Q1: For the HarmBench evaluation - how have the red-teaming methods been applied with the critics in place? …Was the evaluation also done using completions?...
> For the HarmBench benchmark, we followed the original evaluation setup proposed in [1] (see Figure 3 of this work). Specifically, the red-teaming methods like PAP, TAP, or PAIR are applied to the original task prompts (provided in HarmBench). Each red teaming method augments the original task prompts to make it harder for LM actor agents to detect malicious intentions, leading to harmful generation output. The evaluation is then done on the generated completions of the actor agents using the HarmBench AI evaluator. With the critics in place, the evaluation is applied to the actor agent’s completions after it receives the critic feedback and regenerates a new response. Due to the limited space, we described the details of the HarmBench benchmark and references of the red-teaming methods in Appendix D (L996-1003).  We will make the experimental setup and evaluation clearer in our revised paper.
>
> ### Q2: …Could the authors share more details/prompt along with a sample?
> Thanks for your comment. A well-rounded summary is beneficial to ensure all the information of the critic feedback is conveyed to the actor agent. We included the example summary prompt in Appendix F (Listing 3 and 8). Other prompts are also included in this Appendix. We also included more qualitative examples with generated dialogues in the attached PDF in our global response above for your reference. We will include these qualitative samples and explain them in more detail in our revised paper.
>
> ### Q3: I was curious if fine-tuning models (smaller models?) on such generated data would have been a good experiment?...
> Thanks for your suggestion. It would be interesting to conduct finetuning experiments on smaller models using generated data. Currently, there is no available training split in the benchmarks and tasks used in this paper. However, synthetic datasets or relevant crowd-sourced datasets could be considered (e.g. from open-domain Github tasks or other code generation benchmarks).
>
> Furthermore, while using fine-tuned models might be cheaper during inference, they will still be subject to unseen security problems or novel red-teaming malicious prompts. In our global response #3 with fine-tuning methods, the best finetuned model is far from perfect and its performance can be further optimized. Our method INDICT can complement these approaches. Our results show that using INDICT can significantly boost the performance of a fine-tuned model, from 60% (SFT+DPO model) to more than 73%.
>
> *[1] HarmBench: A Standardized Evaluation Framework for Automated Red Teaming and Robust Refusal*
>
> *[2] CodeUltraFeedback: An LLM-as-a-Judge Dataset for Aligning Large Language Models to Coding Preferences*

---

> > ### Author Response · Authors · 2024-08-11
> >
> > Dear Reviewer rV62,
> >
> > We hope our rebuttal response has addressed your concerns about the paper. As the authors-reviewers discussion will end in a few days, please do let us know early if you still have any questions or need further clarification.
> >
> > Regards,

---

> > > ### Comment · Reviewer_rV62 · 2024-08-13
> > >
> > > Thank you for the response -- I have no further questions. These would be good to include in the updated manuscript. I would like to retain my review scores.

---

> > > > ### Author Response · Authors · 2024-08-14
> > > >
> > > > Thank you again for your feedback and response to our rebuttal! We will incorporate all feedback and our discussion into the revised paper.
> > > >
> > > > Regards,

---

### Official Review · Reviewer_SdG3 · 2024-07-08

**Soundness:** 4
**Presentation:** 3
**Contribution:** 2
**Rating:** 6
**Confidence:** 2

**Summary:**

This paper proposes a framework for generating both safe and helpful code. It integrates an internal dialogues of critiques against the given task and the corresponding generated response. It queries external knowledge through relevant code snippets and tools like web search and code interpreter. INDICT is evaluated on multiple tasks across 8 different languages. The results show an advanced level of critiques could significantly improve the code quality.

**Strengths:**

This paper presents a very technically sound system for generating safer code. It improves the safety of code generation with both preemptive and post-hoc feedback, which is quite complete. The authors conduct solid and thorough experiments for their system, which proves significant improvement over existing methods.

**Weaknesses:**

This paper presents a useful and well-designed framework. However, I have some doubts about the novelty of the proposed method. The framework uses actor-critic architectures and includes safety and helpfulness critics, which have been mentioned in previous papers like [1].

In this framework, the critics use data from web searches, Wikipedia, and OpenAI, which raises my concerns about data leakage. For example, it might be easy to find the original web page for a CWE by searching its description when evaluating on the datasets built from CWEs. Using OpenAI as a knowledge base could also have similar issues. Because of this, while the framework performs well on benchmarks, I’m concerned about the generalizability of this framework.

Besides, as the paper mentioned, the code execution could invoke unexpected consequences. It seems the authors do not address this issue, which might require sandboxing execution or other safe execution methods.

[1] Le, H., Wang, Y., Gotmare, A. D., Savarese, S., and Hoi, S. C. H. (2022). Coderl: Mastering code generation through pretrained models and deep reinforcement learning. Advances in Neural Information Processing Systems, 35:21314–21328.

**Questions:**

1. Can you provide more details on how your method differs from existing actor-critic architectures? Specifically, what new contributions does your framework make compared to previous works?
2. How do the safety and helpfulness critics contribute to the overall performance of the framework? Are there any specific examples or case studies you can provide to illustrate their impact?

**Limitations:**

Yes

---

> ### Author Rebuttal · Authors · 2024-08-06
>
> ### Q1: …how your method differs from existing actor-critic architectures?...
>
> We provided a systematic and comprehensive comparison between INDICT and related actor-critic approaches such as CodeRL in our global response#1 above. Compared to existing actor-critic methods, INDICT is different in three major aspects:
> * (1) INDICT aims to optimize both helpfulness and safety awareness in the generated output code. Most of the current actor-critic approaches are designed with a single criterion (such as functional correctness). Simply extending these methods with additional instructions on safety criteria is sub-optimal (see our results with baseline actor-critic methods in our global response#2 above).
> * (2) INDICT integrates critic with knowledge grounding from external tools to create more reliable feedback to the actor agent. Most current methods only use code test results as the only external feedback to improve the quality of output code.
> * (3) To implement (1) and (2), we enhanced the existing actor-critic framework with a multi-critic and tool-enabled collaboration approach. This approach can autonomously generate more reliable and holistic feedback for both the safety and helpfulness of output code generation.
>
> We will include a more detailed comparison in our revised paper.
>
> ### Q2: In this framework, the critics use data from web searches, Wikipedia, and OpenAI, which raises my concerns about data leakage…Because of this, while the framework performs well on benchmarks, I’m concerned about the generalizability of this framework.
> Thanks for raising this concern. We want to highlight that our use of external tools is only for querying relevant information snippets to supplement the critic feedback, not to directly find a solution to a given task. We do not provide any additional information to the critic to generate queries e.g. no information about the related CWEs or security issues, so the critic would have to independently decide what information to choose to search. We conducted an analysis of the generated tool queries by the critic agents and found that on average, for each task, words that are found in queries only overlap with less than 5% of the total words in the original task description. Therefore, the data leakage is very minimal.
>
> Furthermore, from our experiment results (Table 2 in the paper), even without access to external tools, our approach with a multi-critic collaboration system still leads to significant performance gains in base models like CodeLlama or CommandR. When stronger and more recent GPT models are used (e.g. results with GPT4o-mini in our global response#2), the direct generation result is still rather low i.e. models are not heavily exposed to the test data, and more improvement could be done with methods like INDICT. We will include a more detailed discussion in our revised paper.
>
> ### Q3: Besides, as the paper mentioned, the code execution could invoke unexpected consequences…
>
> Thank you for your suggestion. Sandbox environment is a technical engineering feature that we would like to integrate with INDICT. However, the novelty of our method should still be guaranteed. We will describe in more detail the requirement for a sandbox environment for secure code execution in our revised paper.
>
> ### Q4: How do the safety and helpfulness critics contribute to the overall performance of the framework? Are there any specific examples or case studies you can provide to illustrate their impact?
>
> As demonstrated in our ablation experiments in the paper (see Table 3), using individual component critics leads to sub-optimal performance and is not as good as our method with a multi-critic collaboration system. To illustrate the benefits of our approach, we included a qualitative example in the PDF attached to our global response above (see Figure 1).
>
> Compared to the baseline methods, our approach can simultaneously improve both the helpfulness and safety of the output code generation. Specifically, given a relevant information snippet by the safety critic (about the hashing method SHA-256), our actor agent correctly revised the code with a more secure hashing method, avoiding using MD5 hashing and the common security risk CWE-328. At the same time, our generated code is generally more helpful with properly modularized functions implementing supporting features such as input validations. As we noted, this feature has generally emerged in code solutions by collaborative agent systems like CAMEL and INDICT. We will explain the qualitative samples in more detail in our revised paper.

---

> > ### Author Response · Authors · 2024-08-11
> >
> > Dear Reviewer SdG3,
> >
> > We hope our rebuttal response has addressed your concerns about the paper. As the authors-reviewers discussion will end in a few days, please do let us know early if you still have any questions or need further clarification.
> >
> > Regards,

---

> > > ### Comment · Reviewer_SdG3 · 2024-08-13
> > > **Response to authors**
> > >
> > > Thank you for your clarifications. Your response looks good to me. I would suggest adding the data leakage part to your experiments or appendix to resolve audiences' concern. Raise my score.

---

> > > > ### Author Response · Authors · 2024-08-13
> > > >
> > > > Thank you for your response! We very much appreciate your consideration and decision to raise the review score. We will incorporate our rebuttal discussion in detail in the revised paper.
> > > >
> > > > Regards,

---

### Official Review · Reviewer_GLux · 2024-07-11

**Soundness:** 2
**Presentation:** 2
**Contribution:** 2
**Rating:** 5
**Confidence:** 5

**Summary:**

This paper presents a new framework called INDICT that employs two complementary critic agents to improve both the safety and helpfulness of LLM-generated code. Each critic agent is obtained by prompting an LLM with task-specific instructions and knowledge obtained from external tools such as web search and Wikipedia. To better generate knowledge-grounded critics, the authors propose a thought-action-observation process where the critic agent first generates the initial critic without external knowledge, then predicts search keywords or code snippets for knowledge retrieval based on the initial critic, and finally executes the external tools to retrieve the knowledge. The two critic agents can interact with each other by including the critics generated by the other agent in the current turn and past interactions in the prompt. To avoid computation overhead, INDICT also includes a summarization agent to summarize the previous interactions and uses the summary as the context for the next turn. INDICT is evaluated on eight different tasks across eight programming languages from five benchmarks. The results show that INDICT can achieve better results compared with vanilla LLMs such as GPT-4 and Llama-3. The authors also did an ablation study to show the effectiveness of using two critic agents vs. one agent and incorporating critics after the initial code generation vs. after code execution.

**Strengths:**

1. The idea of using two complementary critic agents and using a summarizer to avoid the computation overhead is new and interesting.
2. The proposed framework is evaluated on multiple benchmarks, tasks, and programming languages.
3. The results demonstrated the effectiveness of using two critic agents.

==Post Rebuttal Comment===

The additional results provided in the rebuttal look promising and I believe this work can be strengthened by including those results. However, due to the time limit and also the lack of details in the rebuttal response, I am not able to make a full assessment of the correctness of the results.

**Weaknesses:**

1. My main concern about this work is the weak comparison baselines. In the evaluation, INDICT is only compared with vanilla LLMs rather than advanced and relevant prompting methods such as CodeRL, Self-Correct, and Self-Collaboration. While these prompting methods are initially designed for a single criterion such as code helpfulness, the authors can simply include an additional instruction about the other criterion in the original prompts of these methods to instruct the LLM to give two kinds of critics. This sounds like a more realistic baseline. It would be interesting to see to what extent combining two separate critic agents via the sophisticated interaction proposed by this work outperforms simply mentioning two critic criteria in a SOTA self-critic (or self-refinement) pipeline.
2. The ablation study does not really examine the effectiveness of the novel technical bits proposed in this work. Compared with simply combining two critic agents as in a multi-agent framework, a novel idea in this work is to use a summarizer to avoid computation overhead. However, this summarizer is not evaluated. The authors should compare INDICT vs. INDICT without a summarizer.
3. The thought-action-observation is also an interesting idea that should be evaluated in the ablation study. How much improvement does this process achieve compared with existing RAG and tool-enhanced methods?
4. Does INDICT use zero-shot prompting or few-shot prompting in each step? Based on the prompts in Appendix F, it seems INDICT uses zero-shot prompting in each step. However, this design is not very convincing for certain steps like generating relevant text queries or code snippets. Without few-shot demonstrations, it is hard to restrict the LLM to generate valid outputs. So I was wondering how exactly the authors did that. Also, how did the authors extract search keywords from the LLM response?
5. The justification for using a GPT evaluator is weak. While one or two published papers may have adopted that method, it does not mean that it is a rigorous evaluation method and should be followed broadly by the community. To improve the rigor of evaluation, the authors should consider performing a manual analysis of a small set of samples.
6. The analysis of the experiment results is shallow and does not provide deep insights like why INDICT performs better than other methods and where INDICT fails. The authors should add a failure case analysis.
7. In the related work, the authors should provide a more detailed discussion of how INDICT differs from existing self-critic approaches and multi-agent frameworks. The current discussion is short and handwavy.

**Questions:**

1. How well does INDICT perform compared with the stronger baselines mentioned above?
2. What is the effectiveness of using a summarizer?
3. How much improvement does the thought-action-observation process achieve compared with existing RAG and tool-enhanced methods?
4. Does INDICT use zero-shot prompting or few-shot prompting in each step?

**Limitations:**

The discussion on limitations looks reasonable. It would be helpful to have some quantification on the computation cost.

---

> ### Author Rebuttal · Authors · 2024-08-06
>
> Thank you for your comments! Please refer to our responses below to your questions.
>
> ### Q1: …In the evaluation, INDICT is only compared with vanilla LLMs…
> Following your recommendations, we selected 7 strong baselines from different research lines (see our global response #2 above). We also include a version where additional instructions are given to models to provide both safety and helpfulness critics e.g. instruct models to “focus on both the security and helpfulness of the solution.” For multi-agent methods, we included these instructions in all agents (analyst, tester, etc.) or introduced a new critic agent (as recommended in CAMEL's appendix). Note that for all baseline models, we followed similar generation budgets to fairly compare the results (up to 3 rounds of revision). Our results demonstrate the SoTA performance of INDICT against all the baselines. While we observed the improvement of baseline methods with additional instructions (marked with the suffix ‘+’), their results are sub-optimal. We will include all results and a more detailed analysis in our final revision of the paper.
>
> ### Q2: …The authors should compare INDICT vs. INDICT without a summarizer.
> We conducted a new ablation to evaluate the summarizer (see our global response#3 above). We simply removed the summarizer and let the actor agent receive the full dialogue history. From the results, we noticed the performance degraded to 87% and 72% in safety and helpfulness. This happens probably due to the much longer context of the dialogue history, affecting the actor agent to capture all critic feedback from this history and generate new code. This model variant also incurs more computation due to the long context of the dialogues.  We will include the results and more detailed analysis in our revised paper.
>
> ### Q3: How much improvement does the thought-action-observation process achieve compared with existing RAG and tool-enhanced methods?
> We conducted additional experiments (see our global response#3 above) with two simple variants: (1) RAG-based critics which use the original task description to retrieve relevant knowledge and generate augmented critiques; and (2) tool-based critics which directly use external tools to query “what is wrong with the solution in terms of its <security/functionality>?”; the query output is then treated as a critic feedback. The results show that these variants are inferior to our proposed framework. We found that the queries in these methods are often too vague or ambiguous to search for meaningful information snippets.
>
> ### Q4: Does INDICT use zero-shot prompting or few-shot prompting in each step? …
> INDICT uses zero-shot prompting in each step. We prompt the critic agent to condition the current critique and generate a unique query to obtain more knowledge. We extract the search keywords following our instruction templates e.g. in the form of 'Search[keyword]'. For generating code snippets, we prompt the model similarly but ask the model to wrap the output code in ```. Note that the current prompts in the Appendix (Listing 4 and 5) are for tool-finetuned models like CommandR which automatically generates tool parameters given a tool calling function definition. We will include more prompts for these steps for other standard models in our revised paper.
>
> ### Q5: ...the authors should consider performing a manual analysis of a small set of samples.
> To supplement the GPT-based results in helpfulness, we manually evaluated a small set of 40 samples from CyberSecEval1 (5 random samples per language). From the Table below, we found that even though GPT predicted results are often higher than human evaluation, the observation of INDICT's better performance still stands against the baselines.
>
> | **Baseline** | **Evaluation** | **INDICT wins** | **Draw** | **Baseline wins** |
> |:---:|:---:|:---:|:---:|:---:|
> | **Direct Gen** | Human | 74.8 | 8.5 | 16.7 |
> |  | GPT | 80.2 | 0.3 | 19.5 |
> | **Reflexion+** | Human | 57.1 | 4.2 | 38.7 |
> |  | GPT | 58.4 | 0.0 | 41.6 |
>
> Note that the security metric is done by the code analyzer from [1] which already demonstrated a high level of accuracy in detecting security issues.
>
> ### Q6: …provide deep insights like why INDICT performs better than other methods and where INDICT fails.
> From our additional results (see our global response#2 and #3 above), we noticed that the best baseline models (Self-repair+ or Reflexion+) are as good as an ablation method with basic tool-based critiques where the external tool is vaguely queried: “What is wrong with the current solution?”. This shows that the quality of the critic feedback in the current methods is not good enough, simply focusing on shallow reviews of generated solutions. In INDICT, we enable the model to analyze and choose what information snippets to query and how to integrate this information into their critiques (e.g. queries about an uncommon security term or certain coding best practices). Combined with our multi-critic collaboration approach, INDICT demonstrates strong empirical results against other methods.
>
> For qualitative results on how INDICT performs better and example failure scenarios, please refer to our attached PDF in the global response above. We will include more analysis in our revised paper.
>
> ### Q7: ... how INDICT differs from existing self-critic approaches and multi-agent frameworks.
> Please refer to our global response#1 for a systematic comparison of INDICT and the related work, including the self-critic approaches and multi-agent frameworks. We reviewed each method by the following features: helpfulness or safety-based qualities of generation outputs, execution feedback (execution of output code if applicable), tool-enhanced feedback (access to external tools like web search), multi-critic collaboration (engage multiple LM agents for critic generation) and supervision free (no training data required). We will include the full details of our comparison in the revised paper.

---

> > ### Author Response · Authors · 2024-08-13
> >
> > Dear Reviewer GLux,
> >
> > Thank you again for your reviews and comments on our paper. In our rebuttal, we have tried to address your concerns as much as possible, including your main concern about comparison to baselines (Q1), ablation experiments (Q2 and Q3), and other concerns such as prompting, manual evaluation, comparison to related work, etc. (Q4 to Q7). Please let us know if you still have any questions before the authors-reviewers discussion ends soon.
> >
> > Regards,

---

> > > ### Comment · Reviewer_GLux · 2024-08-14
> > > **Response to the rebuttal**
> > >
> > > I wanted to thank the authors for their responses. The new experiment results with stronger baselines and ablation studies are helpful. The authors should include those results in their paper and provide more interpretation & elaboration on them.
> > >
> > > Overall, the results look promising. I raised my score to borderline accept to acknowledge the effort from the reviewers. But I have to admit that I am not able to make a full assessment of the new results as in a regular review due to the lack of details on the results. I feel these new results require another round of careful reviews since these results are not simply the addition of a secondary experiment but rather completely new results with new baselines, etc.

---

> > > > ### Author Response · Authors · 2024-08-14
> > > >
> > > > Thank you very much for your consideration and revision of the review score! Due to the limited character constraint in rebuttal, some information might not be fully explained. We will elaborate in detail the new results and other discussion in our revised paper.
> > > >
> > > > Regards,

---

> ### Author Response · Authors · 2024-08-11
>
> Dear Reviewer GLux,
>
> We hope our rebuttal response has addressed your concerns about the paper. As the authors-reviewers discussion will end in a few days, please do let us know early if you still have any questions or need further clarification.
>
> Regards,

---

### Author Rebuttal · Authors · 2024-08-06

We thank the reviewers for providing insightful comments on our paper. Please refer to this global response for our high-level answers to the common concerns. For more detailed explanations and analysis, please refer to the corresponding threads of individual reviewers.

### 1. Comparison with related baselines and our contributions
In the table below, we provided a more comprehensive and systematic comparison of our method to related work. We compared INDICT and related methods from 3 directions: self-refine/self-critic, multi-agent, and finetuning. Compared to these methods, INDICT is a more well-rounded generation framework with the following contributions: (1) integrates code execution-based feedback and enhances them with external knowledge, (2) targets both helpfulness and safety of output code, and (3) facilitates an interactive and supervision-free multi-agent collaboration framework. Our additional experiment results (see the next response) showcase the efficacy of INDICT.

| Method | Helpfulness | Safety | Execution feedback | Tool-enhanced | Multi-critic collab | Supervision free |
|---|:---:|:---:|:---:|:---:|:---:|:---:|
| **_Self-refine approach_** |  |  |  |  |  |  |
| CodeT, AlphaCode, MBR-Exec | ✅ |  | ✅ |  |  | ✅ |
| Self-correct, ILF | ✅ |  |  |  |  | ✅ |
| CodeRL, Self-edit | ✅ |  | ✅ |  |  |  |
| Self-repair, Self-debug, Reflexion | ✅ |  | ✅ |  |  | ✅ |
| **_Multi-agent approach_** |  |  |  |  |  |  |
| Self-collaboration, AgentCoder | ✅ |  | ✅ |  |  | ✅ |
| CAMEL | ✅ |  |  |  |  | ✅ |
| ChatDev, Self-org Agents | ✅ |  | ✅ |  | ✅ (?) | ✅ |
| MetaGPT, AgentVerse | ✅ |  | ✅ | ✅ |  | ✅ |
| **_Finetuning approach_** |  |  |  |  |  |  |
| CodeUltraFeedback, StableAlignment | ✅ | ✅ |  |  | ✅  |  |
| SafeCoder | ✅ | ✅ | ✅ |  |  |  |
| **INDICT** | **✅** | **✅** | **✅** | **✅** | **✅** | **✅** |

### 2. Additional experiment results to compare with stronger related baselines
From the above table, we selected representative baselines and evaluated them on a validation test split (random samples of 20% of the CyberSecEval benchmark). With GPT4o-mini as the base model, we adapted the baselines in their original implementation and also extended them with additional instructions (detailed criteria of safety and helpfulness). We marked these enhanced baselines with the suffix ‘+’. We observed that INDICT can lead to SoTA performance by both security and helpfulness (more than 90% and 81% respectively). Strong baselines like Reflexion+ and CAMEL+ improve with additional critic instructions but are not as strong as INDICT.
| **Method** | **Safety** | **Helpfulness** | **S+H** |
|---|:---:|:---:|:---:|
| Direct Gen | 78.2 | 50.0 | 64.1 |
| INDICT | **90.9** | **81.4** | **86.1** |
| **_Sefl-refine methods_** |  |  |  |
| Self-debug | 80.0 | 52.7 | 66.3 |
| Self-debug+ | 79.7 | 53.9 | 66.8 |
| Self-correct | 80.7 | 59.7 | 70.2 |
| Self-correct+ | 86.7 | 68.5 | 77.6 |
| Self-repair | 83.7 | 69.6 | 76.6 |
| Self-repair+ | 86.6 | 70.9 | 78.8 |
| Reflexion | 83.3 | 68.5 | 75.9 |
| Reflexion+ | 86.9 | 69.6 | 78.2 |
| **_LM agent methods_** |  |  |  |
| Self-collab | 78.7 | 52.3 | 65.5 |
| Self-collab+ | 79.1 | 66.2 | 72.7 |
| CAMEL | 81.6 | 63.7 | 72.6 |
| CAMEL+ | 82.6 | 70.2 | 76.4 |

We also conducted experiments to compare INDICT with finetuning-based methods. Using CodeLlama-7b-instruct as the base model, CodeUltraFeedback finetunes the model on a large-scale dataset with annotations of code preferences. We observe that the best model (SFT + DPO finetuning) can improve the results by both safety and helpfulness but not as good as INDICT. As INDICT can complement finetuning-based methods, we applied INDICT with the best CodeUltraFeedback model to achieve even further performance gains (from 60% and 63% to 73%).

| **INDICT vs. finetuning methods** | **Safety** | **Helpfulness** | **S+H** |
|---|:---:|:---:|:---:|
| Direct Gen | 56.3 | 50.0 | 53.2 |
| INDICT | 65.3 | 62.1 | 63.7 |
| CodeUltraFeedback (SFT) | 58.5 | 49.9 | 54.2 |
| CodeUltraFeedback (DPO) | 62.7 | 56.0 | 59.3 |
| CodeUltraFeedback (SFT+DPO) | 63.9 | 57.9 | 60.9 |
| CodeUltraFeedback (SFT+DPO) +INDICT | **74.9** | **72.4** | **73.7** |

### 3. Additional ablation results to demonstrate the effectiveness of the proposed components
Using GPT4o-mini as the base model, we conducted additional ablation experiments with different variants of INDICT: (1) one simply using a single critic agent for both safety and helpfulness; (2) one without using a critic summarizer and maintaining a full dialogue history of critiques in the critic context; (3) ones replacing the thought-action-observation critic generation with RAG or tool-based generation: (3a) RAG uses the original task description to retrieve relevant knowledge and generate grounded critics, and (3b) tool-based method uses web search/Wikipedia and a query “what is wrong with the solution in terms of its <security/functionality>?” and query output is treated as a critique. The result table below shows that our proposed INDICT obtains the optimal performance, with a more well-rounded performance in both safety and helpfulness.
| **Ablation methods** | **Safety** | **Helpfulness** | **S+H** |
|---|:---:|:---:|:---:|
| INDICT (full) | **90.9** | **81.4** | **86.1** |
| - one critic for both criteria | 87.3 | 76.4 | 81.9 |
| - no critic summary | 87.9 | 72.2 | 80.1 |
| - RAG-based critics | 87.9 | 74.4 | 81.1 |
| - tool-based critics | 85.5 | 72.7 | 79.1 |

### 4. Qualitative analysis to demonstrate the benefits of our methods on generated code and analyze failure cases
To address the concerns about the benefits of INDICT and where INDICT may fail, we demonstrated some qualitative analysis in the attached PDF document. In Figure 1, we show that given a code generation task, INDICT can generate code that is more secure and robust than strong baselines (Direct generation, Reflexion, and CAMEL). In Figure 2, we illustrate cases where INDICT may fail due to nontrivial errors.

---

### Author Response · Authors · 2024-08-09
**Authors-reviewers discussion**

Dear Reviewers,

As the authors-reviewers discussion period ends on 13 Aug, please let us know if you have any more concerns or clarifications needed to be addressed. Appreciate if you can let us know early so we can facilitate more timely discussion.

Regards,

---

> ### Author Response · Authors · 2024-08-13
>
> Dear Reviewers,
>
> We hope our rebuttal has addressed your concerns sufficiently. Before the authors-reviewers discussion period ends on 13 Aug, please let us know if any additional information or clarification is still needed to improve the paper.
>
> Regards,

---

### Decision · Program_Chairs · 2024-09-25

**Decision:**

Accept (poster)

**Comment:**

This paper presents a technique for improving the quality and safety of code generation systems. By using two separate critics (one for helpfulness, one for safety) with access to each other and endowing them with access to information (via search tools) a multi-turn code generation system is shown to produce more accurate and safer code on a range of benchmarks.

All reviewers agreed that the approach is novel and useful. The evaluations, including those provided in the rebuttal, are extensive and show sufficient evidence about the benefits of this approach. As reviewers mention the cost-related tradeoffs are not well-discussed or evaluated. In particular the critics have additional cost and represent additional "thinking" (compute) done on each problem. Equi-cost comparisons of the various methods, ablations, and baselines would resolve the question about the extent to which INDICT's double-critic approach works because of its design or because of the additional compute available to the approach.

Despite the above, all the reviewers believe that there is no sufficient reason to reject this work and hence recommend its acceptance.

The authors are kindly asked to incorporate the results discussed in the rebuttal and -- at minimum -- a discussion about the costs of INDICT to be included.